# Dissipation and Distribution of Picarbutrazox Residue Following Spraying with an Unmanned Aerial Vehicle on Chinese Cabbage (*Brassica campestris* var. *pekinensis*)

**DOI:** 10.3390/molecules26185671

**Published:** 2021-09-18

**Authors:** Chang Jo Kim, Won Tae Jeong, Kee Sung Kyung, Hee-Dong Lee, Danbi Kim, Ho Sung Song, Younkoo Kang, Hyun Ho Noh

**Affiliations:** 1Residual Agrochemical Assessment Division, Department of Agro-Food Safety and Crop Protection, National Institute of Agricultural Sciences, Wanju 55365, Korea; rlackdwh1@gmail.com (C.J.K.); shewaspretty@korea.kr (W.T.J.); yihd@korea.kr (H.-D.L.); danbi6334@korea.kr (D.K.); 2Department of Environmental and Biological Chemistry, College of Agriculture, Life and Environment Science, Chungbuk National University, Cheongju 28644, Korea; kskyung@cbnu.ac.kr; 3Disaster Prevention Engineering Division, Department of Agricultural Engineering, National Institute of Agricultural Science, Wanju 55365, Korea; ddeeuuxx@korea.kr (H.S.S.); ykk0977@korea.kr (Y.K.); 4Upland Mechanization Team, Department of Agricultural Engineering, National Institute of Agricultural Science, Wanju 55365, Korea

**Keywords:** pesticide residue, Chinese cabbage, liquid chromatography, tandem mass spectrometry, QuEChERS, unmanned aerial vehicle, multicopter, picarbutrazox, spraying condition

## Abstract

We assessed the residual distribution and temporal trend of picarbutrazox sprayed by agricultural multicopters on Chinese cabbage and considered fortification levels and flying speeds. In plot 2, 14 days after the last spraying, the residues decreased by ~91.3% compared with those in the samples on day 0. The residues in the crops decreased by ~40.8% of the initial concentration owing to growth (dilution effect) and by ~50.6% after excluding the dilution effect. As the flight speed increased, picarbutrazox residues decreased (*p* < 0.05, least significant deviation [LSD]). At 2 m s^−1^ flight speed, the residual distribution differed from the dilution rate of the spraying solution. The average range of picarbutrazox residues at all sampling points was 0.007 to 0.486, below the limit of quantitation −0.395, 0.005–0.316, and 0.005–0.289 mg kg^−1^ in plots 1, 2, 3, and 4, respectively, showing significant differences (*p* < 0.05, LSD). These results indicated that the residual distribution of picarbutrazox sprayed by using a multicopter on the Chinese cabbages was not uniform. However, the residues were less than the maximum residue limit in all plots. Accordingly, picarbutrazox was considered to have a low risk to human health if it was sprayed on cabbage according to the recommended spraying conditions.

## 1. Introduction

Because of the increasing human population globally, the production of food based on agricultural products should be increased by approximately 25–70% by 2025 compared with that in 2014 [1]. However, it will be difficult to produce high-quality agricultural products without a rescue plan owing to the deterioration of the environment and limited resources [1]. Additionally, loss of productivity and/or unforeseen disruptions such as the coronavirus disease 2019 (COVID-19) pandemic hinder the import of crops because of export limitations at the origins [2]. Despite the substantial efforts needed to improve agricultural productivity, almost all agricultural industries in the developing countries rely on conventional agriculture, which results in labor shortages [3]. This not only lowers the returns to farmers, but also creates considerable differences between the demand and supply of agricultural products [4]. With 14.9% of the total population over 65 years of age in 2019, Korea has become an aged society—a serious situation in the agricultural scenario. Moreover, the elderly population is expected to enter an ultra-aged society by 2025 (20.3%), and labor shortage is expected to accelerate [5]. These problems are not limited to South Korea as China and North America are also experiencing labor shortages at different stages, such as the spraying of pesticides [6].

Pesticide spraying using manned aircrafts has emerged as an alternative solution. However, although this approach can be applied to large-scale cultivated lands, its application is limited in East Asia, where the land is small and irregular. To solve this problem, pesticide spraying using unmanned helicopters was started in Japan in the 1990s; as the multicopter industry developed rapidly, multicopters were implemented even in China [3]. Pesticide spraying using multicopters has become popular mainly in the cultivation of rice, corn, and cotton, as well as crops such as apple and tea [7,8,9,10]. In Korea, it is also applied to crops such as Chinese cabbage, potato, and green onions. When unmanned aerial vehicles (UAVs) are used, the amount of chemicals required is less than that for other spraying methods, which can reduce the cost of agricultural materials and decrease the probability of pesticides being leached into the groundwater [11]. Moreover, the agricultural multicopter is 100- and 30-fold more efficient in crop protection activities than human resources and other general spraying practices, respectively; thus, it can reduce time and labor requirements. Furthermore, it can minimize pesticide exposure in workers and damage to crops by virtue of its approach to fields via the air [12]. Therefore, aerial spraying has numerous advantages; however, manned and unmanned aerial spraying raise concerns over potential wind-induced drift [13,14,15]. Drifted pesticides may deposit residues on crops in the surrounding farms, and they could be transported to other arable lands and rivers, with potentially adverse effects on ecosystems [16]. Therefore, research on technologies or management strategies to minimize wind-induced drift should be carried out.

Although a multicopter for spraying pesticides is a spray tool distinct from the traditional ones, the sprayed pesticides could be lost and decomposed based on the environmental conditions and the chemical properties of the pesticides, with some remaining in the soil and crops [17]. As the residue’s characteristics may vary with the nozzle type employed for spraying and the application dosage [18], residue distribution of pesticides sprayed using agricultural multicopters on crops should be evaluated.

In Korea, agricultural multicopters are allowed to spray pesticides on a variety of crops. Among vegetables, root vegetables, whose underground parts are consumed after foliar spraying, are less likely to be exposed to pesticides [19]. Meanwhile, in cereals such as rice, most of the remaining pesticides are lost in the threshing process, whereas in beans and corn, several residual pesticides are removed as they are peeled [20]. Residual pesticide distribution is likely to vary with the systemic properties and characteristics of the cultivated area [21]. However, pesticides are likely to remain in vegetables such as Chinese cabbage, and research on the properties of pesticides should be prioritized [22]. Among them, Chinese cabbage, the test crop in this study, is the most consumed in Korea [23]. Therefore, the evaluation of pesticide residues in this crop should be prioritized over those in other crops.

The use of pesticides for crop protection allows the production of high-quality agricultural products and storage for a long period, but it has potentially adverse effects on the environment and humans [24]. As the pesticide sprayed remains on crops and is consumed by humans [25], acceptable daily intake (ADI) and maximum residue limits (MRL) are defined for each country to manage the pesticides. While pesticides are sprayed on crops, large amounts of pesticides can remain in the soil depending on their chemical properties and on the soil properties, adversely affecting beneficial animals and microorganisms [24]. Furthermore, residual pesticides that transit to the water system may eventually return to the human body because of bioconcentration [26,27,28].

Therefore, pesticide residues in crops, mobility in soil, and toxicity, which require toxicology and soil mobility experiments, must be evaluated. The consequences of toxicity may not significantly vary with the spraying and environmental conditions, but pesticide residues may vary. Because residual pesticides can vary among crops depending on the spraying and environmental conditions, it is important to evaluate residual patterns according to spraying.

Here, we used picarbutrazox as the test pesticide. It is a fungicide that can be classified as carbamate and pyridine and is mainly used to treat *Phytophthora* blight and downy mildew in Korea. It can be applied using agricultural multicopters on Chinese cabbage. Picarbutrazox is widely used in Korea and, as it passes the MRL, is considered to have low environmental effects; furthermore, it has relatively low toxicity. The total radioactive residue (TRR) of TZ-1E, a major metabolite of picarbutrazox, is more than 10% of the total residue. The toxicity of TZ-1E is similar to that of picarbutrazox, and its residual concentration may be considerably higher than that of the parent compound. Furthermore, data regarding the fate of the pesticide picarbutrazox are limited. Therefore, residues in plants must be defined and managed as the sum of the residues of picarbutrazox and TZ-1E [29].

In this study, we aimed to evaluate the safety of spraying picarbutrazox on Chinese cabbage when considering environmental and spraying conditions and to investigate residual distribution and time trends.

## 2. Results and Discussion

### 2.1. Meteorological Data

On day 1 of spraying, the wind speed was 2.4–3.5 m s^−1^ from the southwest or south-southwest. In contrast, there was no wind effect on day 2 of spraying. On day 3 of spraying, the wind speed was 1.2 m s^−1^ from the north-northeast in plot 1. The temperature, humidity, wind speed, and direction are listed in Appendix A.

### 2.2. Limit of Quantitation, Reproducibility and Recovery

The limit of quantitation (LOQ) of the analytical method was 5 µg kg^−1^, that is, the signal-to-noise ratio of more than 10, showing valid reproducibility. In the reproducibility test results, the coefficient of variation (CV) of peak area, height, and retention time were 3.47%, 3.93%, and 0.39%, respectively, for picarbutrazox and TZ-1E on the chromatogram at LOQ and 10LOQ. The recovered picarbutrazox and TZ-1E in the Chinese cabbage ranged from 85.1% to 114.3% (Figure 1), indicating that this method is suitable for pesticide residue analysis.

### 2.3. Time-Dependent Residual Distribution

The residues in plot 2, where pesticide was sprayed according to the pre-harvest interval (PHI) and the recommended spraying conditions for multicopters in Korea, were 0.265, 0.206, 0.188, 0.170, 0.051, 0.044, and 0.023 mg kg^−1^ on days 0, 1, 3, 5, 7, 10, and 14, respectively (Figure 2). The persistence ratio of the residue 14 days after the initial residue was approximately 8.7%, indicating that there was a significant time-dependent decrease (*p* < 0.05, least significant deviation (LSD)). Based on this, the equation of the regression curve was used to calculate the half-life of picarbutrazox in Chinese cabbage (Figure 2).

Generally, the dissipation of pesticide residues in crops depends on the formulation; the physicochemical properties (vapor pressure and solubility); the climate conditions (temperature, humidity, sunlight, and rainfall); the crop characteristics (shape, genus, and species); the location; the frequency; and the application dose [30,31,32,33], with the dilution effect of pesticide residues caused by crop growth being the major factor [19]. Noh et al. [34] reported that the residues of picarbutrazox in shallots from 0 to 14 days after the last spray with multicopters decreased from 1.521 to 0.020 mg kg^−1^ in a time-dependent manner. Here, the time-dependent residual dissipation of picarbutrazox in the Chinese cabbage was similar to that in the shallots, but the residues in the shallots were higher than those in the Chinese cabbage. This can be attributed to the larger exposed surface area-to-mass ratio in shallots than in Chinese cabbage and the dilution effect, as Chinese cabbage is heavier than shallots [35]. Bae et al. [36] sprayed boscalid and fludioxonil on leafy vegetables, such as spinach, Chinese cabbage, tatsoi, and Chinese vegetables and found that the amount of pesticide residue in spinach was 2.5–3.0 times higher than that in the other leafy vegetables. This result was attributed to the growth velocity, leaf surface, life-form, and specific surface area characteristics of the test crops.

### 2.4. Dissipation and Dilution Effect

The half-life of pesticides varies with the type of crop and the properties of the pesticides, and it ranges from 0.6 to 29 days [19]. The half-life of pesticides in vegetables is 0.5–17 days and that of fungicides is 1.5–17 days [19]. Here, the half-life of picarbutrazox in the Chinese cabbage was 4 days, consistent with the established range [19]. Chinese cabbage is a relatively fast-growing crop (Figure 3), and the pesticide is thought to have a short half-life owing to the dilution effect. However, the residues may be reduced by pesticide decomposition. Therefore, the residue dilution effects caused by crop growth and by excluding crop growth were calculated.

The residues of picarbutrazox under the dilution effect caused by crop growth decreased from 0.265 to 0.157 mg kg^−1^. The equation of dissipation curve obtained was *y* = 0.2780 × *e*^−0.0409x^ (r^2^ = 0.9668) (Figure 4), and the persistence ratio of residue at 14 days after the initial residue was approximately 59.2%. The half-life was approximately 17 days, as determined using a first-order kinetic model; it was longer than the biological half-life. This result showed that the concentration of residual pesticides decreases as Chinese cabbage grows; a similar result was also reported through a linear regression curve for the dissipation of pesticide residues sprayed on cabbage [36]. Excluding the dilution effect, the residues were 0.265 and 0.131 mg kg^−1^ at 0 and 14 days after the last spraying, respectively, and the equation for the dissipation curve was *y* = 0.2257 × *e*^−0.0515x^ (r^2^ = 0.6206) (Figure 4). The persistence ratio of the residue 14 days after the first day was approximately 49.4%, with a half-life of approximately 14 d. This was longer than the biological half-life and shorter than the residual half-life based on the dilution effect. In conclusion, picarbutrazox residue in Chinese cabbage was diluted and reduced by crop growth, but it was also lost by decomposition, with higher losses than those by dilution alone.

### 2.5. Residues Owing to Treatment

According to the results in Figure 5, the average residue in plots 1 and 2, with the same dilution rate (16-fold) and different flight speeds (2 and 3 m s^−1^), was 0.113 and 0.074 mg kg^−1^, respectively. Whereas, that in plots 3 and 4, with the same dilution rate (32-fold) and different flight speeds (2 and 3 m s^−1^), was 0.089 and 0.066 mg kg^−1^, respectively.

Adhesion is inversely related to droplet size, droplet velocity, and incidence angle on the leaf [37], and if the spraying volume increases, the amount of pesticide applied to the crops and adhesion increases [3]. However, after a threshold, the amount of adhesion may not increase. Here, 800 and 600 mL of pesticides were sprayed at speeds of 2 and 3 m s^−1^, respectively. Therefore, the residues in the plot where the spraying dose was large owing to the slow speed were higher than those in the other cases.

Considering the same flight speed of 2 m s^−1^, the average residues in plots 1 and 3, with 16- and 32-fold dilutions, were 0.113 and 0.089 mg kg^−1^, respectively, with the residual amount being high in plot 1, which had a high concentration of spraying solution. However, despite the concentration being twice that in plot 3, the actual residue increased only 1.27-fold. At 3 m s^−1^, the average residues in plots 2 and 4 were 16- and 32-fold diluted, respectively, compared with those at 2 m s^−1^ spraying (*p* < 0.05). This shows that the spraying volume affects the residue, but this effect can be offset by adjusting the UAV flight speed.

### 2.6. Distribution of Residues

Although the degree of pesticide spraying with UAVs varies depending on crop characteristics such as the canopy, it is generally difficult to uniformly spread pesticides owing to the physical characteristics of UAVs and wind effects [3]. Besides the environmental conditions, the main factors affecting aerial spraying are altitude and speed, as these factors form an eddy caused by downwash and lead to uneven spraying [38]. Here, the average residual amounts in the Chinese cabbage were 0.007–0.486, <LOQ–0.395, 0.005–0.316, and 0.005–0.289 mg kg^−1^ in plots 1, 2, 3, and 4, respectively, with residues on each plot being significantly different among the sampling points, indicating that the residual distribution of picarbutrazox sprayed on the Chinese cabbage with an agricultural multicopter was not uniform (Appendix A). Wen et al. [39] reported that spraying solutions through nozzles tends to cross spray and concentrate the residue in the center; in fact, when an octacopter with four nozzles is used for spraying, the amount of spray in the center was high. However, in this study, although the residue in the center tended to be concentrated, the residual amount in the center was high. Alternatively, the residues in the samples at the beginning and at the end of spraying may be excessive. If pesticide spraying and flight are not performed simultaneously and the flight starts after the start of spraying, the residue may be excessive at the start point. Even at the end of spraying, if the spraying and flight are not stopped simultaneously, the residue may be overestimated or undervalued [34]. Moreover, it takes time for a multicopter to implement a set speed after starting a flight, and because the multicopter flight speed may be below the set speed at the start of spraying, relatively more pesticide is likely to be sprayed at this point.

In conclusion, we recommend the adoption of our method as a good agricultural practice. This method would also reduce the MRL compared with traditional methods. Nevertheless, to minimize the effect of the agricultural environment caused by drift, spraying methods should be developed considering drift reduction. To this end, it is necessary to develop drift guard nozzles, physical drift reduction technologies, and adjuvants that would minimize drift.

## 3. Materials and Methods

### 3.1. Test Pesticide and Crop

The test pesticide was a 10% suspension concentrate of picarbutrazox. In Korea, good agricultural practice (GAP) recommends an application dose of 1.6 L 10 a^−1^ and allows spraying up to three times until 7 days before harvest [40]. For the test crop, Chinese cabbage (*Brassica campestris* var. *Pekinensis*), the MRL set by the Ministry of Food and Drug Safety in Korea is 2 mg kg^−1^ (the sum of picarbutrazox and its metabolite TZ-1E).

### 3.2. Field Trials

A hexacopter (OneTop A1; Topflight Co., Ltd., Gwangju, South Korea) equipped with a DG11002 nozzle (Spraying System^®^, Chicago, IL, USA), which reduces drift [41] and has an effective spraying width of 4 m, was used. This method can spray the pesticide to seven rows of Chinese cabbage in one direction. The discharge rate was set to 1200 mL min^−1^ to match the recommended spraying rate of Korean PHI (1.6 L 10 a^−1^). An automatic weather station equipped with a CR1000X series datalogger combined with EE181 (temperature and humidity) and 03002 (wind speed and direction) sensors (Campbell Scientific, Logan, UT, USA) was set to measure these weather conditions at 5 s intervals during spray.

There were four experimental plots that received the following combinations of dilution and flight speed treatments: plot 1, 16-fold and 2 m s^−1^; plot 2, 16-fold and 3 m s^−1^ (recommended treatment); plot 3, 32-fold and 2 m s^−1^; and plot 4, 32-fold and 3 m s^−1^. Each area of the test plots (4 m × 100 m, W × L) was sprayed three times until 7 days before harvest for the worst-case scenario, which was expected to result in the maximum residue according to the Korean GAP.

The Chinese cabbages in plot 2, sprayed according to the Korean PHI, were randomly collected on days 0 (after 3 h), 1, 3, 5, 7, 10, and 14 after the last spray to survey residual dissipation. To investigate Chinese cabbage growth, the samples were weighed on each sampling day. To investigate the residual distribution of picarbutrazox, 42 samples were collected in each plot (Figure 6) on day 7 after the last spray, that is, the due date for pre-harvest. The collected samples were blended with dry ice after removing the roots and old leaves and were stored in a freezer at −20 °C.

### 3.3. Stock Solution and Matrix-Matched Standard

We dissolved 20.24 and 20.08 mg of picarbutrazox and its metabolite TZ-1E standard, respectively, in 20 mL of acetonitrile to prepare a stock solution of 1000 mg L^−1^. Each stock solution was mixed with acetonitrile at 100, 50, 25, 10, and 5 mg L^−1^. Using a 5 mg L^−1^ mixture standard, standard solutions of 2, 10, 20, 40, and 100 µg L^−1^ were obtained for creating the calibration curve. Electrospray ionization in mass spectrometry can have a matrix effect during atmospheric pressure ionization. Therefore, matrix-matched standards of 1, 5, 10, 20, and 50 µg L^−1^ made of untreated sample extraction solution were used to offset the matrix effect [42].

### 3.4. Sample Preparation

Acetonitrile was used as a water-soluble organic solvent to extract picarbutrazox and TZ-1E from the Chinese cabbage using the QuEChERS method [43]. To remove interferential matter in samples, d-SPE was used for purification because it contains primary secondary amine (PSA), which can purify fatty acids, organic acids, and graphitized carbon black (GCB), which can remove pigments [44,45]. Information related to the reagents and materials used in this study are presented in the “Reagents and materials” section in the Appendix A.

Ten grams of Chinese cabbage was weighed into a 50-mL conical flask; acetonitrile was added to the mixture, which was then shaken at 1300 rpm for 10 min. The QuEChERS EN kit, containing 4 g of MgSO_4_, 1 g of NaCl, 1 g of trisodium citrate dihydrate, and 0.5 g disodium hydrogen citrate sesquihydrate, was added to the flask and shaken at 1300 rpm for 5 min. The samples were centrifuged at 2332× *g* for 5 min, and the supernatant was added to a d-SPE tube containing 150 mg of MgSO_4_, 25 mg of PSA, and 2.5 mg of GCB. These were centrifuged at 16,128× *g* for 5 min after vortexing for 30 s. The supernatants were passed through a syringe filter. The obtained solutions were diluted two-fold with acetonitrile for matrix matching and then analyzed by LC-MS/MS. The corresponding results are presented in the “Optimization of mass spectrometry” section in the Appendix A. In addition, LOQ, reproducibility, and recovery were determined according to the method described in the Appendix A.

### 3.5. Linearity of Calibration

The matrix-matched standards of 1, 5, 10, 20, and 50 µg mL^−1^ were analyzed using an optimized analytical method, and the linear equations of picarbutrazox and TZ-1E were *y* = 1.134 × 10^8^*x* − 1.520 × 10^3^ and *y* = 3.331 × 10^6^*x* − 4.221 × 10^2^, respectively, with the target compounds exhibiting excellent linearity, r^2^ > 0.999.

### 3.6. Biological Half-Life and Dilution Effect on Pesticide Residue

Various models have been developed for biological half-life (*t*_1/2_; days) in plants. In general, if the correlation coefficient value (r^2^) obtained using the time-dependent residue (n ≥ 5) was 0.7, the half-lives were calculated using a first-order kinetic model [38]. Therefore, it was applied in this study (Equations (1) and (2)).
(1)$$C_{t}=C_{0}e^{-kt}$$
(2)$$t_{1/2}=\frac{\ln\left(2\right)}{k}$$
where *C_t_* (mg kg^−1^) is the concentration of pesticide at time *t* (day), *C*_0_ (mg kg^−1^) is the concentration of pesticides at time *t* = 0, and *k* (day^−1^) is the dissipation rate constant reflecting the degradation potential of the pesticide.

The major factor in determining pesticide residue is the growth-based dilution effect (Equation (3)) caused by crop growth [46]. Therefore, the dilution effect on picarbutrazox residues in Chinese cabbage and the amount of change in absolute pesticides excluding the dilution effect (Equation (4)) were investigated [47].
(3)$$C_{dt}=\frac{C_{0}\times W_{0}}{W_{t}}$$
(4)$$C_{edt}=C_{0}-\left(C_{dt}-C_{t}\right)$$
where *C_dt_* and *C_edt_* (mg kg^−1^) are the concentrations by including and excluding the dilution effect, respectively, after the elapsed time *t*; *C*_0_ and *C_t_* (mg kg^−1^) are the concentrations on day 0 and after the elapsed time *t*, respectively, and *W*_0_ and *W_t_* (g) are the weights of the sample on day 0 and that after the elapsed time *t*, respectively.

### 3.7. Statistical Analysis

Statistical analyses were performed using the one-way analysis of variance (ANOVA) in IBM SPSS Statistics 23 (IBM Corp., Armonk, NY, USA), and Duncan’s multiple range test was performed at *p* < 0.05 to determine the LSD between sample means. To compare the recovery results, the fortification levels were used as dependent variables in the statistical analysis. To compare the residual concentration in the samples, sampling date, sampling points, and plots were considered as dependent factors.

## Figures and Tables

**Figure 1 molecules-26-05671-f001:**
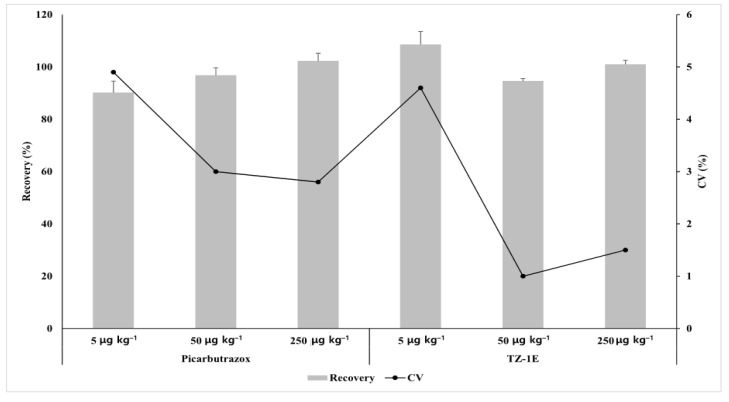
Average recovery with standard deviation (n = 3) and coefficient of variation (CV) of picarbutrazox and TZ-1E in Chinese cabbage at different fortification levels.

**Figure 2 molecules-26-05671-f002:**
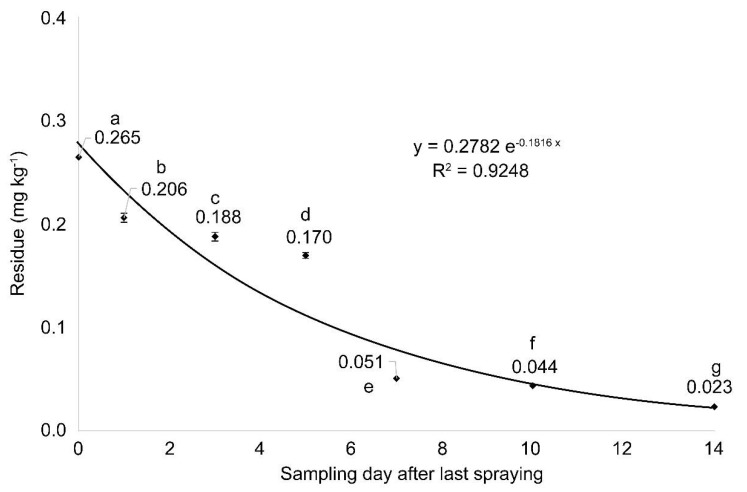
Average residual concentration with standard deviation (n = 3) of picarbutrazox sprayed using a multicopter on Chinese cabbage throughout the sampling period in plot 2. Different letters indicate significant difference at *p* < 0.05 by least significant deviation.

**Figure 3 molecules-26-05671-f003:**
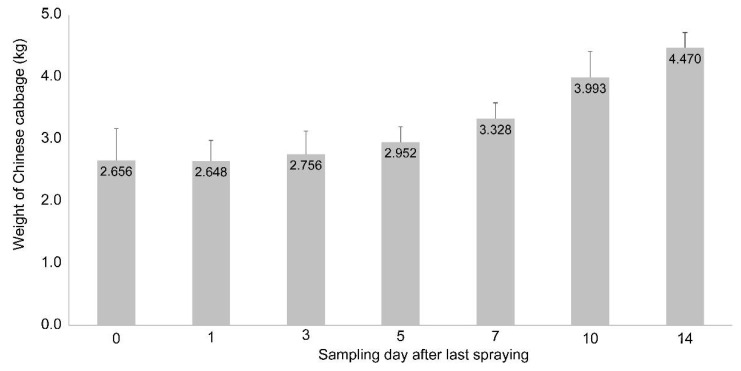
Weight with standard deviation (n = 10) of Chinese cabbage throughout the sampling period in plot 2.

**Figure 4 molecules-26-05671-f004:**
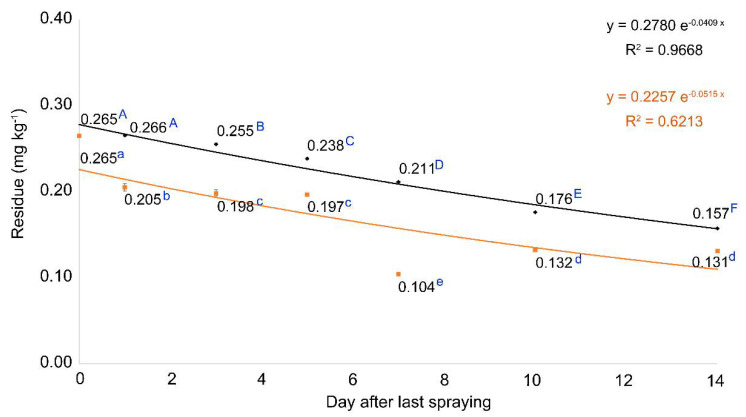
Dissipation of picarbutrazox residue with standard deviation (n = 3) over time in Chinese cabbage, sprayed using a multicopter in plot 2. Black and yellow represent the residue in Chinese cabbage excluding and only considering the growth dilution effect, respectively. Different letters in the dissipation curve indicate significant differences at *p* < 0.05 by least significant deviation.

**Figure 5 molecules-26-05671-f005:**
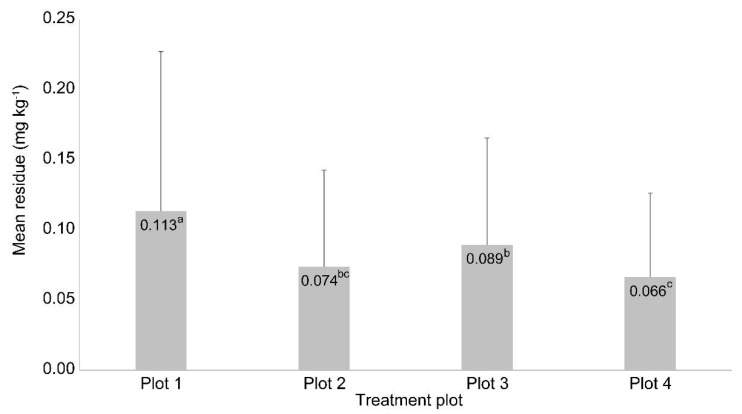
Residues of picarbutrazox with standard deviation in Chinese cabbage at different dilution rates and multicopter speeds. Plots 1 and 2, 16-fold dilution, 2 and 3 m s^−1^ multicopter speeds, respectively; plots 3 and 4, 32-fold dilution, 2 and 3 m s^−1^ multicopter speeds, respectively. Different letters indicate significant differences at *p* < 0.05 based on least significant deviation.

**Figure 6 molecules-26-05671-f006:**
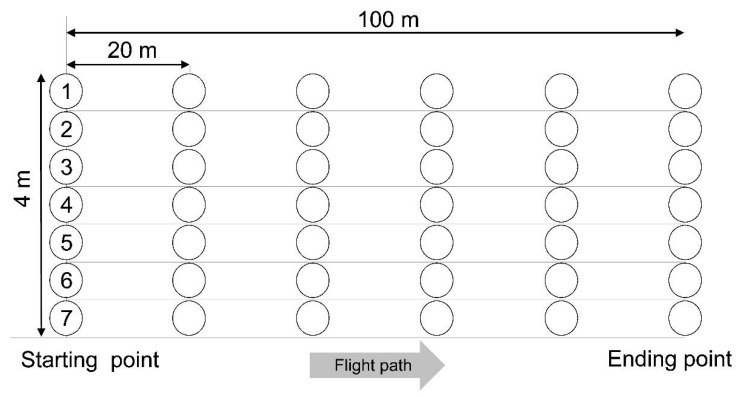
Sampling points in a test plot for the investigation of residual distribution on pre-harvest day.

## Data Availability

Not applicable.

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
