# Peer review of "Dissipation and Distribution of Picarbutrazox Residue Following Spraying with an Unmanned Aerial Vehicle on Chinese Cabbage (Brassica campestris var. pekinensis)"

_molecules, 2021, doi:10.3390/molecules26185671_

Round 1

Reviewer 1 Report

The manuscript entitled “Dissipation and distribution of picarbutrazox residue following spraying with unmanned aerial vehicle onto Chinese cabbage (Brassica campestris var. pekinensis)” describes the fate of the pesticide picarbutrazox after its spraying onto cabbage crops by means of multicopters. The effects of some spraying factors (pesticide concentration and multicopters flight-speed) were investigated, showing that there would be an optimized combination of these parameters.

This is an interesting work about a topic which for sure deserves investigation, because of the possible harmful effects due to the presence of pesticides (and, more in general, micropollutants) in the environment.

Comments about the manuscript are the following.

Introduction:

  • line #56: “spraying3”, please correct the citation format.
  • line #89. “has an adverse effect”. I would suggest using the plural “has possible adverse effects”.
  • lines #94-95. Please, explain better the way by which pesticides can return to humans because of biomagnification.
  • lines #103-111. The authors should add more information about the use and the environmental fate of picarbutrazox (biological, chemical, photochemical degradation; soil adsorption) based on the already published literature.

Results and discussion:

  • line #123. I would suggest “The LoQ of the analytical method”.
  • line #137. How could the authors calculate the half-life time of picarbutrazox from the square Pearson correlation coefficient (r2) of the regression curve?
  • lines #203-215. Authors say that the mean residue of picarbutrazox found on cabbage depends on the multicopters flight-speed and pesticide dosage. They found statistically significant differences between the mean residues of each plot, as per the least significant deviation from ANOVA. However, the standard deviation bars reported in Fig.6 are very large suggesting, at a first glance, that the difference between the mean values does not seem significant. Have the authors used, before ANOVA, some hypothesis test (e.g., T-test and/or F-test) to verify these differences as well?
  • line #222. How could the residue value be negative? Please, explain.

Material and methods.

  • line #306. Please, use the scientific notation for expressing numbers in the relevant equations.
  • line #314. Eq.2. This equation can be a source of confusion for the reader. Although it is correctly derived from Eq.1, probably the authors used regression methods to compute the value of the dissipation rate constant (as in Fig.5). If it is so, please remove Eq.2.
  • line #324. I would suggest adding, if possible, a brief explanation about how to derive Eq.5.

Reviewer 2 Report

The current study assessed the residual distribution and temporal trend of picarbutrazox sprayed with agricultural multicopters on Chinese cabbage considering fortification levels and flying speeds. I am not familiar with multicopters and drones. In my opinion, the core technology for current study was only LC-MS/MS that was used for measuring picarbutrazox content in plants. However, I think the results are interesting and recommend it for publication after minor revisions. I have only some suggestions below.

(1) Picarbutrazox is widely used in Korea, is below normal toxicity, and is considered to have a low environmental impact. Why authors focused on this type of pesticides? In my opinion, picarbutrazox is safe.

(2) I suggest authors to introduce the method of LC-MS/MS that is the core technology in the current study, not show it in the supporting information.

(3) I do not think there is a relationship of weight of Chinese cabbage with the sampling day after the last spraying.

(4) I suggest authors to discuss the advantages of spraying with unmanned aerial vehicle.

Round 2

Reviewer 1 Report

The manuscript has been revised by authors, with no substantial changes of the results. The manuscript in the present form could be still improved.

Minor comments:

  • Abstract, line #26: “-0.007” would be “0.007”.
  • Abstract, line #29: “spraying using” should be “spraying by using”.
  • line #48. In “agricultural industries in the developing countries, rely on conventional” avoid the comma.
  • line #68: “spray methods” should be “spraying methods”.
  • lines #81-82: Instead of “spray tool distinct from traditional tools”, I would suggest adjusting the sentence as “spray tool distinct from the traditional ones” to avoid repetition.
  • line #96. Instead of “Among them”, “Among vegetables” would be clearer for the reader.
  • lines #116-126. In this paragraph, authors should add a sentence explaining that very few data about the fate of the pesticide picarbutrazox are known to date.
  • line #192 and related. As authors say, we do not know much about the fate of picarbutrazox in the environment, if it is more biodegradable, photolabile or even unstable towards chemical hydrolysis. A question arises here as a consequence: is it correct to define the dissipation of the pesticide (that by excluding the dilution effect) as ‘biological’? Or would it be better just to refer to ‘dissipation’?
  • lines #194-195. This is very general and can be misleading to set definite intervals for pesticides half-life in this way. It would be better to replace it with some examples of pesticides or at least to describe that in the cited work ([19]) they have found those intervals.
  • lines #232-236. It would be helpful for the reader to add the pesticide fraction remaining on cabbage as well. Related to this, was the on-cabbage starting concentration of the pesticide equal among plots?
  • Supplementary Information: please, write the right title of the manuscript.
  • Supplementary Information, Optimization of Mass Spectrometry. Please, add more details about the measurements: type of chromatograph used, type of mass detector, column details (length, internal diameter, spheres size), retention times of the analytes.

Author Response

Point 1: line #26: “-0.007” would be “0.007”.

Response 1: We thank you for pointing this out. We have made the necessary revision.

Point 2: line #29: “spraying using” should be “spraying by using”.

Response 2: We thank you for the comment. We have made the necessary revision.

Point 3: line #48. In “agricultural industries in the developing countries, rely on conventional” avoid the comma.

Response 3: We thank you for pointing this out. We have made the necessary revision.

Point 4: line #68: “spray methods” should be “spraying methods”.

Response 4: We thank you for the comment. We have made the necessary revision.

Point 5: lines #81-82: Instead of “spray tool distinct from traditional tools”, I would suggest adjusting the sentence as “spray tool distinct from the traditional ones” to avoid repetition.

Response 5: We thank you for the comment. We have made the necessary revision.

Point 6: line #96. Instead of “Among them”, “Among vegetables” would be clearer for the reader.

Response 6: We thank you for the comment. We have made the necessary revision.

Point 7: lines #116-126. In this paragraph, authors should add a sentence explaining that very few data about the fate of the pesticide picarbutrazox are known to date.

Response 7: We thank you for the comment. We have made the necessary revision.

Point 8: line #192 and related. As authors say, we do not know much about the fate of picarbutrazox in the environment, if it is more biodegradable, photolabile or even unstable towards chemical hydrolysis. A question arises here as a consequence: is it correct to define the dissipation of the pesticide (that by excluding the dilution effect) as ‘biological’? Or would it be better just to refer to ‘dissipation’?

Response 8: We have replaced “Biological half-life” with “Dissipation” in the subheading.

Point 9: lines #194-195. This is very general and can be misleading to set definite intervals for pesticides half-life in this way. It would be better to replace it with some examples of pesticides or at least to describe that in the cited work ([19]) they have found those intervals.

Response 9: We thank you for the comment. We have made the necessary revision.

Point 10: lines #232-236. It would be helpful for the reader to add the pesticide fraction remaining on cabbage as well. Related to this, was the on-cabbage starting concentration of the pesticide equal among plots?

Response 10: Different spraying methods were used for different plots. Owing to aerial spraying, the initial residual amount is bound to be different. Although the residual rate cannot be calculated without examining the initial concentration, we believe that readers will be aware of residual reduction because the residual amounts at harvest time were all different, and we have presented a specific plot reduction pattern.

Point 11: Supplementary Information: please, write the right title of the manuscript.

Supplementary Information, Optimization of Mass Spectrometry. Please, add more details about the measurements: type of chromatograph used, type of mass detector, column details (length, internal diameter, spheres size), retention times of the analytes.

Response 11: We thank you for the comment. We have made the necessary revision.
